# The Arabic Version of the Adult Eating Behavior Questionnaire among Saudi Population: Translation and Validation

**DOI:** 10.3390/nu14214705

**Published:** 2022-11-07

**Authors:** Mona A. Alruwaitaa, Aldanah Alshathri, Lama Alajllan, Norah Alshahrani, Wejdan Alotaibi, Iffat Elbarazi, Madhawi M. Aldhwayan

**Affiliations:** 1Community Health Sciences, College of Applied Medical Sciences, King Saud University, Riyadh 11433, Saudi Arabia; 2Institute of Public Health, College of Medicine and Health Sciences, United Arab Emirates University, Al Ain 15551, United Arab Emirates

**Keywords:** appetite, adult, Arabic, appetitive traits, validation, AEBQ, eating behavior, eating habit, questionnaire

## Abstract

Inherited individual differences in eating behaviors known as “appetitive traits” can be measured using the Adult Eating Behavior Questionnaire (AEBQ). The AEBQ can be used to assess individuals that require intervention regarding their weight, eating habits, and for the identification of eating disorders. Arabic eating behavior assessment tools are few. This study, therefore, aimed to translate and validate the AEBQ in Arabic language (AEBQ-Ar) and to confirm the factor structure while assessing the internal consistency of all subscales. Participants completed the AEBQ-Ar and reported their sociodemographic data online. Exploratory factor analysis (EFA) was used and internal reliability was assessed using Cronbach’s α. Correlations between AEBQ-Ar subscales and body mass index (BMI) were done using Pearson’s correlation. A sample of 596 adults, mean age of 35.61 ± 12.85 years, was recruited from Saudi Arabia. The 6-factor structure was the best model, excluding emotional under- eating subscale and merging enjoyment of food and food responsiveness subscales. Internal consistency was acceptable for all subscales (Cronbach’s α = 0.89–0.66). Emotional over- eating was positively associated with BMI, and slowness in eating was negatively associated with BMI. The AEBQ-Ar with 6-subscales appears to be a valid and reliable psychometric questionnaire to assess appetitive traits in Arabic speakers.

## 1. Introduction

Obesity over the past 2 decades has increased and continues to be a public health concern across the globe that demonstrates association with health behaviors and health outcomes, such as obesity, diabetes, hypertension, and cancer [1]. Eating behaviors play a crucial role in human growth and development, as well as the prevention of lifelong chronic diseases such as obesity [2,3]. Weight gain and obesity are significantly influenced by eating behavior traits [4,5]. 

Nevertheless, eating behavior is a broad term that includes different elements such as food choice and motives, feeding practices, dieting, and eating-related problems, including obesity, eating disorders, and feeding disorders [6]. Additionally, they are becoming more widely acknowledged as crucial elements of a healthy diet that not only address diet quality but also establish the context and motivation behind food intake [7]. Inherited individual differences in eating behaviors known as “appetitive traits” can trigger individuals into overeating or undereating through internal and external stimuli [8], resulting in body weight differences [9]. Individual differences in appetitive traits, including high responsiveness and sensitivity to food cues such as eating in response to the sight, smell, or taste of palatable foods and emotion-based eating, all have consistently been associated with excess food intake and higher body weight [10,11]. However, appetitive traits are not only asssociated with higher body weight, some appetitive traits such as food fussiness, also known as “picky eating” or “selective/neophobic eating” and satiety responsiveness appear to be risk factors for avoidant/restrictive food intake disorder (ARFID) [8]. 

Most valid eating behavior questionnaires among adults focus on overeating and/or obesity risk or eating disorders such as the Three Factor Eating Questionnaire (TFEQ) and the Dutch Eating Behavior Questionnaire (DEBQ) [12,13,14]. The Adult Eating Behavior Questionnaire (AEBQ) has been developed with 8 subscales to cover both; the food approach (Hunger, Food Responsiveness, Emotional Overeating, and Food Enjoyment) and avoidance appetitive traits (Satiety Responsiveness, Food Fussiness, Emotional Undereating and Slowness in Eating) [15]. The questionnaire was adapted from the Child Eating Behavior Questionnaire (CEBQ) to form a psychometrically valid self-reported questionnaire. The AEBQ can be used to report a need for intervention to individuals to control their weight and eating habits by providing precise feedback on managing appetitive trait responses. AEBQ may also help identify subjects at risk of weight gain or eating disorders to provide proper prevention [15]. Following the development of the AEBQ, several studies have examined its validity in different populations (Australia, United States, United Kingdome, Bulgaria, Poland, Portugal, China, Mexico, and Canada). Of these studies, several were done on adolescents (United Kingdom [16], Poland [17], Portugal [18]) others were implemented on young adults (Australia [9], China [8]). Most studies supported the findings that AEBQ is a reliable and valid measure of appetitive traits in adults [9,19,20]. In addition, a superior model fit with a seven-factor model excluding Hunger subscale was found in contrast to the original eight-factor structure [19,20,21].

To the best of our knowledge, the AEBQ is the most comprehensive tool for measuring adults’ self-reported appetitive traits and to date it has not been translated previously into the Arabic language. Furthermore, given that Arabic is the official language of 25 countries, translating and validating the AEBQ in Arabic is necessary. Thus, our aims were first to translate the AEBQ into Arabic; second, to evaluate the validity of the translated version of the questionnaire on a Saudi Arabian population; third, to confirm the factor structure of the Arabic version of AEBQ (AEBQ-Arabic); and finally, to evaluate the internal consistency among all subscales.

## 2. Materials and Methods

This was a cross-sectional study. The study included two main steps: translation process, followed by validation process. Before starting, permission was taken from the original authors of the AEBQ to translate into the Arabic language and to evaluate its validity on the Saudi Arabian population. 

### 2.1. Translation Process

The World Health Organization (WHO) translation, and adaptation process was followed to create the Arabic version of the AEBQ [22]. 

#### 2.1.1. Forward Translation 

The team included seven native Arabic speakers fluent in English (five clinical dietitians, a senior clinical dietitian as independent translators, and an expert in methodology and public health). Each member of the team created a first draft. After that, all versions were compiled into one unified version by choosing the most appropriate translation, followed by a one-day focus group with a native Arabic expert familiar with the discipline and all seven team members to finalize the forward translation version of the questionnaire.

#### 2.1.2. Back-Translation

Back translation was carried out by an independent bilingual native English speaker fluent in Arabic and medical terms. The AEBQ was translated back from Arabic into English without viewing the original version of the questionnaire. A critical review was performed to compare the back-translation and the original version, highlighting the terms that needed adjustments. A second meeting was held with the expert to discuss linguistic errors and decide whether to change or keep the same due to language/cultural differences. Six items had minor changes to match the meaning of the terms in the English language. Two independent Arabic language experts did another linguistic review, and nineteen items had minor grammar modifications. Further, a detailed focus group meeting was done to discuss linguistic changes, and the new version of AEBQ was approved for pilot testing.

#### 2.1.3. Pilot Testing

Interviews with a sample of 20 adult participants with equal numbers of males and females with different educational levels were conducted. Participant ages ranged from 18 to 60 years old, while the education level varied between primary, secondary, high school, diploma, bachelor’s degree, Master’s, or Ph. D holders. For each item, participants were asked if the statement was clear, confusing, easy to answer, and relevant to the purpose of the questionnaire. Pilot testing was done through 1:1 interview or phone calls when an interview was not possible, with a duration that ranged between 6–31 min, and notes were taken regarding any confusion or misunderstanding of any item.

Two comments on item 26 of the questionnaire: “I eat more and more slowly during the course of a meal” showed that the statement of the item was not understood as going slower, but rather eating very slowly in one pattern as participants found it similar to item 29 “I eat slowly”. Other frequently repeated comments were the similarity in the vocabulary used, such as annoyed, upset, angry, worried, and anxious, which resulted in some questions being repeated. Two participants were confused regarding item 9 “If I miss a meal, I get irritable” and whether the statement meant getting irritable from being hungry or just missing the mealtime.

#### 2.1.4. Final Version

According to pilot testing, few changes were made to the final version of the questionnaire. First, for words and terms that were frequently reported as confusing and difficult to recognize (annoyed, upset, angry, worried, and anxious), definitions were obtained from a glossary of psychiatric terms and were added next to the questions that include the terms [23]. Second, in item 26, we added the word (deceleration) between brackets to ensure it is understood as going slower during mealtime. 

#### 2.1.5. Content Validity and Experts’ Evaluation

An assessment was done with a panel of 11 experts in different specialties (clinical nutrition, lifestyle medicine, epidemiology and public health, and Arabic language professionals) [24,25]. The experts examined whether the final version was relevant to the topic of the questionnaire or not according to pre-defined relevance scores (1 = not relevant, 2 = unable to assess relevance, 3 = relevant but needs minor alteration, 4 = relevant and succinct) [24]. The scores 3 and 4 were recognized as 1 (relevant) and scores of 1 and 2 as 0 (not relevant). Content validity was measured in different formulas, including item-level content validity index (I-CVI), scale-level content validity index based on the average method (S-CVI/Ave), and scale-level content validity index based on the agreement method (S-CVI/UA) [25]. I-CVI of ten or more experts should be no less than 0.78, S-CVI/Ave of excellent level should be 0.9 or higher, and S-CVI-UA minimum acceptable level is 0.8 [26,27].

### 2.2. Validation Process

The distributed questionnaire started with sociodemographic details (gender, age, education level, nationality, marital status, employment status, family size (Households size), income, and region of living. The participants subjectively reported that body weight and height, and body mass index (BMI) (kg/m^2^) was calculated based on them. The AEBQ used a five-point Likert scale, and the questionnaire compromised of 35-items that were part of 8 subscales. Those subscales included four food approach subscales to assess Hunger (H), which accounts for five items, Food Responsiveness (FR) with four items, Emotional Over-Eating (EOE) with five items, and Enjoyment of Food (EF) with three items [15]. Four food avoidance subscales were used to assess Satiety Responsiveness (SR) with four items, Emotional Under-Eating (EUE) with five items, Food Fussiness (FF) with five items, and Slowness in Eating (SE) with four items [15].

#### 2.2.1. Sample Size Calculation

The recommended sample size for validating a questionnaire is ten participants per item (ratio 10:1), and we needed 350 participants [28]. Moreover, a priori sample size calculation for Structural Equation Modelling (SEM) technique to run Confirmatory Factor Analysis (CFA) was used and showed that a minimum sample size of 177 is required to run CFA [29]. The minimum required sample size to run our analysis was 527 participants to be included in the study.

A total of 596 participants responded to the questionnaire and two participants chose not to complete the questionnaire, while 18 were excluded from the final analysis due to incomplete or incorrect data. The final sample included in the analysis was 576 participants.

#### 2.2.2. Participants

The study was carried out between February 2021 and June 2022. Participants were adults aged 18 years and above and Arabic speakers. The questionnaire was distributed through online platforms such as WhatsApp and Twitter, and none of the questionnaires was completed before participants provided their consent. 

#### 2.2.3. Statistical Analysis

The mean and standard deviation were used to describe continuous variables and the categorically measured variables were described with frequencies and percentages. Histogram and the Kolmogrove–Smirnove test were applied to test the statistical normality assumption and Levene’s test was used to test the homogeneity of variance statistical assumption. Cronbach’s alpha test was used to assess the internal consistency of the questionnaire and its subscale scores, the corrected item-total and item-total if the item is deleted analysis of internal consistency was also used to assess the internal reliability/internal consistency. Exploratory Factor Analysis (EFA) using Principal Components Analysis (PCA) with Parallel Analysis (PA) test were applied to participants measured indicators of the AEBQ- Ar. Closeness-To-Undimensionality (UniCo, M-IREAL, and ECV tests) were used to assess undimensionality of AEBQ-Ar indicators and its overall subscale scores [30]. Sampling adequacy for EFA test was assessed with the Kaplan–Meyer–Olkins (KMO) test and the suitability of the factor analysis for AEBQ-Ar inventory was assessed with Bartellets and determinant index tests. Variance inflation index and the tolerance tests were used to assess the collinearity between the measured indicators of AEBQ-Ar. Pearson’s correlation test (*r*) was used to assess the correlations between the measured concepts. Because social status and poverty is multi-faceted, non-linear factor analysis (i.e., categorical factor analysis) was used to reduce participants measured sociodemographic, educational, occupational level, households’ size, and income factors into a standardized (z-score) socioeconomic status index (SESi) score and its reliability was assessed via categorical Cronbach’s alpha test and indicator loadings to the latent socioeconomic state index. SESi was used as a proxy independent predictor variable to characterize participants’ socioeconomic state in the multivariate analyses [31]. SPSS IBM (IBM Corp., Armonk, NY, USA) statistical analysis program was used for the statistical data analysis and FACTOR stand-alone statistical program used for performing the parallel analysis test under the exploratory factor analysis context [30]. The statistical Alpha significance level was considered at the 0.050 level.

#### 2.2.4. Ethical Considerations 

The study was approved by the ethics committee board of human and social studies at King Saud University, Saudi Arabia (IRB: 836-21). All participants provided informed consent to participate in this research.

## 3. Results and Discussion

### 3.1. Content Validity

Content validation produced an overall validity index of over (0.78) except for item 6 (I often notice my stomach rumbling), I-CVI, was (0.73) as shown in Appendix A Table A1, as it was explained that the stomach could make many sounds even if the person was not physically hungry. For that, we specified the word (rumbling) to be between brackets to drag attention. S-CVI/Ave produced an acceptable level of (0.95), and S-CVI/UA produced a level of (0.5). Although S-CVI/UA is considered low, it is difficult to reach a high level with such a high number of experts [27].

### 3.2. Characteristic of Study Participants 

Five hundred and seventy-six people residing in Saudi Arabia had enrolled themselves electively in the study and completed and returned the online survey. Table 1 shows the sociodemographic characteristics of the study population. The majority of the sample (72%) were females. The mean ± standard deviation (SD) age was 35.61 ± 12.85 years. The vast majority of the sample (84.9%) lived in the Central provinces of Saudi Arabia, and (15.1%) were from other provinces of Saudi Arabia. 

### 3.3. Reliability Analysis

Analysis findings in Table 2 showed that AEBQ-Ar 35-items had adequate internal consistency with Cronbach’s α = 0.72. All subscales of the AEBQ-Ar (Hunger, Food responsiveness, Emotional over-eating, Enjoyment of food, Satiety response, Emotional under-eating, Food fussiness, and Slowness in eating) had substantial internal consistency with Cronbach’s α for each of them ≥0.71 approximately. However, hunger and satiety response subscales showed relatively small internal consistency when read by Saudis, Cronbach’s α = 0.66. The corrected item-total score correlations analysis was also considered for the thirty-five items. As shown in Table 3, items measuring emotional under-eating (5 items) had very low corrected item-total score correlations (each had corrected item-total correlations <0.10 points). Three items from the food fussiness subscale (FF 12, FF 19, FF 24) and one item from the slowness in eating subscale (SE 14) had poor corrected item-total correlations. Items of EUE beside those latter three food fussiness items were recommended to be deleted by the analysis program to enhance the internal consistency of the questionnaire.

### 3.4. Factor Analysis 

Descriptive analysis and ascending mean rankings for participants’ perceptions of AEBQ-Ar are listed in Appendix A Table A2. The initial EFA showed the adequacy of the sample size for PCA procedure as evidenced by adequate KMO index (KMO = 0.849), and the Bartelettes sphericity Chi-squared test (χ^2^ (378) = 6354.8, *p* < 0.001) was statistically significant indicating the invertibility of the correlation matrix between questionnaire items with a Determinant Index <0.00001 when the 35 items were considered in one analysis model. However, EUE (five items) resulted in initial extracted variances below 0.2 points with equal cross-loadings to multiple factors. Other items (H 6, FR 13, FR 17) had very low initial extracted variance and swayed under other factors less related to their main construct. These eight items were excluded one at each step while repeating EFA iterations. The final resulted factor analysis in Table 4 showed six latent factors that can be extracted from the correlation’s matrix between the remaining 27-items. Parallel analysis test had agreed with the other tests of a possible number of latent factors (Mean average partial test and scree-cassilith plot) that there were between 4–6 latent meaningful, simple, and theoretically sound latent factors that may be extracted from these 27 items. The tests of Closeness-To-Unidimensionality had all agreed that AEBQ-Ar is not essentially a unidimensional factor and as such, a total score from all these 27-items may not characterize a whole upper-order latent eating behavior score and the consideration of each latent subscale factors is rather more favored as a distinct construct. Unidimensional congruence index (UniCo = 0.683) explained common variance index (ECV = 0.678) and mean item residual absolute loadings index (M-IREAL = 0.252). Moreover, the tests of closeness-to-unidimensionality were also applied to the AEBQ-Ar 6-subscales mean scores to assess whether they may converge on a unidimensional score, and the resulted findings indicated that these subscales might not comprise a unidimensional score either. The final 6-factor analysis was accepted, and the factor solution was rotated with Promax method to allow these subscales to correlate with each other as expected theoretically. Results of factor solution showed that the five items EOE had coalesced under one latent factor that we named (emotional over/under eating). Other items that measured EF had coalesced under the second latent factor that we named (Enjoyment of food). Items of hunger had also loaded saliently to the third latent factor (Hunger), and the 4-items of SE had coalesced and loaded significantly to the fourth factor (Slowness in eating), likewise, all items of FF had also loaded to one latent factor with substantive loadings (>0.50) to their factor (Food fussiness) except for item FF 24, and the items of SR had meaningfully and significantly loaded to their factor (Satiety response). All the items had loaded to their intended latent factors saliently (>0.35) and few items had swayed under other factors but meaningfully too (items: FF 24, FR 22, and FR 33) as they characterized enjoyment of food rather than food responsiveness or food fussiness. Mean scores of these 6-latent factors by averaging the items comprising each latent factor according to the yielded factor analysis solution are listed in Table 5 and shown in Figure 1. Items that were excluded from the factor analysis were also removed from these mean modified subscale scores.

Table 6 displays Bivariate Pearson’s correlations between the AEBQ-Ar subscales and participants characteristics. Food approach scales correlated positively together, mean perceived hunger (H) score had correlated positively with their food enjoyment (FE) mean score (*r* = 0.461, *p* < 0.010). Mean perceived hunger score had also converged significantly and positively with emotional over-eating (EOE) score (*r* = 0.391, *p* < 0.010). Mean perceived food enjoyment had correlated significantly positively with EOE score (*r* = 0.425, *p* < 0.010). As hypothesized, BMI had correlated significantly and positively with one of the food approach scales, EOE mean score (*r* = 0.155, *p* < 0.010). From another side, one of the food avoidance scales, mean SE score had correlated negatively and significantly with BMI and socioeconomic state index scores (*r* = −0.155, *p* < 0.010), (*r* = −0.136, *p* < 0.010), respectively. The socioeconomic status index (SESi) had converged significantly and positively with age (*r* = 0.464, *p* < 0.001).

The present study aimed to translate and evaluate the validity of the Arabic version of the AEBQ on a Saudi Arabian population, to confirm the factor structure, and to evaluate the internal consistency among all subscales. The findings from this study show that AEBQ-Ar is a valid and reliable measurement tool of appetitive traits in a Saudi Arabian population. Reliability estimates exhibit high and acceptable values, which further support the internal consistency of the questionnaire. Items with low corrected item-total score correlations show a poor contribution to the whole questionnaire; thus, they were suggested to be deleted to enhance the internal consistency of the questionnaire.

Our findings did not confirm the use of a 7-factor model (eliminating Hunger subscale) nor the original 8-factor model; instead, the AEBQ-Ar found to be valid to use with a new 6-factor model including 27-items. In AEBQ-Ar, we eliminated EUE subscale (with its five items), Food responsiveness subscale with part of its items (FR 13, FR 17), and one item from Hunger subscale (H 6). A newly merged subscale was made, named Enjoyment of Food (EF), which contains the items of Enjoyment of food subscale, the remaining items from Food responsiveness subscale, and one item from Food fussiness subscale. The new EF subscale contains six items (EF 1, EF 3, EF 4, FF 24, FR 22, and FR 33). The new factor structure by the Arabic version of AEBQ is distinctive compared to other versions of the AEBQ. The 7-factor model (eliminating Hunger subscale) was repeatedly confirmed by other studies validating AEBQ in different populations (Mexico [20], Poland [17], UK [16], Canada [19], and Bulgaria [21]). The original 8-factor model was adapted by (China [8], Australia [9], and the original development study [15]). On the contrary, AEBQ was validated among Portuguese adolescents and yielded 5-factor model with Food responsiveness and Enjoyment of Food combined into one factor, as appearing in our results [18].

There are multiple proposed reasons for excluding EUE factor from AEBQ-Ar. Saudi participants had difficulty differentiating between the items of the two subscales (EUE and EOE). In fact, their questions seem to represent the same idea in the Arabic language. During pilot testing, many participants commented that these questions seemed to be repeated and confusing.

Due to the relatively low reliability, when EUE factor was deleted, the whole questionnaire had a higher value of reliability and internal consistency. During the identification process of the factor structure of the questionnaire, EUE items had a low value of extracted variances and cross-loading into multiple factors, resulting in high residual error loading and adding extra collinearity with items of EOE factor. Other deleted items (H 6, FR 13, FR 17) also had low values of extracted variance and loaded between multiple factors that were less related to the main construct. During experts’ evaluation, item H 6 had relatively low I-CVI, as they pointed out that the stomach can make many sounds other than when feeling hungry.

Our findings partially showed a positive association between BMI and food approach subscales and a negative association with food avoidance subscales as initially hypothesized. BMI correlated positively with EOE, a food approach subscale, and negatively with SE, a food avoidance subscale. These findings align with other studies of the AEBQ. The Mexican population showed a positive correlation with EOE and a negative correlation with SE [20], and the Canadians showed a positive correlation with EOE [19]. Results from Bulgaria showed a positive correlation with EOE [21], and finally China showed a negative correlation with SE [8]. 

This study encountered several limitations, including subjective reporting of weight and height parameters which raises recall and social desirability bias, and the unequal gender distribution among the sample [32,33,34]. Study strengths included the high number of participants, engagement of different specialties during the translation process and experts’ evaluation (Arabic language professionals, medicine, public health professionals and nutrition experts). Future work would consider testing the validity of AEBQ-Ar in clinical settings, such as obesity clinics. 

## 4. Conclusions

The present study suggests that a 27-item, 6-subscale AEBQ would be a convenient questionnaire for assessing appetitive traits in Saudi, Arabic speakers. The findings from this study suggest associations between BMI with food approach and food avoidance scales, but there is a need to explore these associations further in the future. The AEBQ-Ar will enable conducting population-based studies on appetitive traits and obesity risk. Future studies are needed to confirm these findings on different subgroups to allow more diversity. The final version of the AEBQ in Arabic language will be available from the corresponding author (Aldhwayan, M.M.) upon reasonable request.

## Figures and Tables

**Figure 1 nutrients-14-04705-f001:**
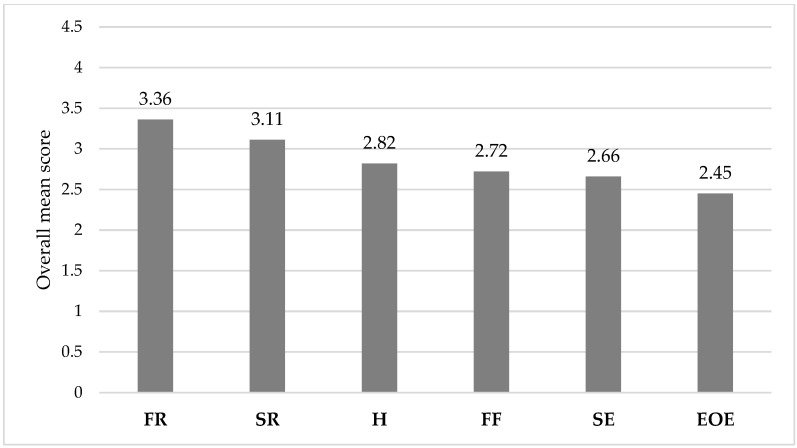
Participants’ overall mean AEBQ subscale scores. AEBQ: Adult Eating Behavior Questionnaire; FR: Food Responsiveness; SR: Satiety Response; H: Hunger; FF: Food Fussiness; SE: Slowness in Eating; EOE: Emotional Over-Eating.

**Table 1 nutrients-14-04705-t001:** Sociodemographic characteristics of study participants, *n* = 576.

	Frequency	Percentage
Sex		
Male	161	28
Female	415	72
Age (years), mean (SD)		35.61 (12.85)
Age group		
20–30 years	278	48.3
31–40 years	117	20.3
41–50 years	94	16.3
51–60 years	67	11.6
≥61 years	20	3.5
Body weight (kg), mean (SD)		70.21 (18.78)
Body Height (centimeters), mean (SD)		163.10 (9.10)
Body Mass Index, mean (SD)		26.29 (6.23)
Body Mass Index Level		
Underweight	63	10.9
Normal	210	36.5
Over-weight	177	30.7
Obese Class I	91	15.8
Obese Class II	35	6.1
Marital state		
Never married	248	43.1
Ever married	328	56.9
Educational Level		
High school or less	66	11.5
Diploma Degree	63	10.9
University Degree	373	64.8
Higher studies	74	12.8
Employment state		
Student	97	16.8
Unemployed	134	23.3
Retired	76	13.2
Employed	269	46.7
Households size (family members), mean (SD)	Mdn = 6	5.86 (2.21)
Family size (Households size)		
1–3 members	75	13
4–6 members	279	48.4
7–10 members	209	36.3
≥11 members	13	2.3
Households monthly Income Level		
Less than 5000 SAR	43	7.5
Between 5000 to 9999 SAR	108	18.8
Between 10,000 to 14,999 SAR	132	22.9
Between 15,000 to 20,000 SAR	97	16.8
More than 20,000 SAR	196	34
Living region		
Central region	489	84.9
Northern region	17	3
Southern region	11	1.9
Western region	33	5.7
Eastern region	26	4.5

SD, standard deviation; SAR, Saudi Riyal.

**Table 2 nutrients-14-04705-t002:** The reliability analysis of AEBQ-Ar and its subscales.

	Number of Items	Cronbach’s α
Adult Eating Behavior Questionnaire-Arabic (AEBQ-Ar)	35	0.720
Food Approach Subscales		
Hunger (H)	5	0.660
Food Responsiveness (FR)	4	0.714
Emotional Over-Eating (EOE)	5	0.895
Enjoyment of Food (EF)	3	0.759
Food Avoidance Subscales		
Satiety Response (SR)	4	0.660
Emotional Under-Eating (EUE)	5	0.856
Food Fussiness (FF)	5	0.716
Slowness in Eating (SE)	4	0.818

**Table 3 nutrients-14-04705-t003:** Item-total statistics of AEBQ-Ar.

		Corrected Item-Total Correlation	Cronbach’s α If Item Deleted
EF 1	I love food	0.148	0.715
EF 3	I enjoy eating	0.131	0.716
EF 4	I look forward to mealtimes	0.315	0.706
EOE 5	I eat more when I’m annoyed	0.303	0.706
EOE 8	I eat more when I’m worried	0.290	0.707
EOE 10	I eat more when I’m upset	0.325	0.705
EOE 16	I eat more when I’m anxious	0.249	0.710
EOE 21	I eat more when I’m angry	0.204	0.712
EUE 15	I eat less when I’m worried	0.008	0.725
EUE 18	I eat less when I’m angry	0.156	0.716
EUE 20	I eat less when I’m upset	−0.008	0.725
EUE 27	I eat less when I’m annoyed	0.102	0.719
EUE 35	I eat less when I’m anxious	0.090	0.719
FF 2	I often decide that I don’t like a food, before tasting it	0.254	0.709
FF 7	I refuse new foods at first	0.266	0.709
FF 12	I enjoy tasting new foods	0.026	0.721
FF 19	I am interested in tasting new food I haven’t tasted before	−0.079	0.728
FF 24	I enjoy a wide variety of foods	−0.095	0.725
FR 13	I often feel hungry when I’m with someone who is eating	0.318	0.706
FR 17	Given the choice, I would eat most if the time	0.364	0.702
FR 22	I am always thinking about food	0.389	0.702
FR 33	When I see or smell food that I like, it makes me want to eat	0.357	0.705
H 6	I often notice my stomach rumbling	0.296	0.707
H 9	If I miss a meal I get irritable	0.352	0.703
H 28	I often feel so hungry that I have to eat something right away	0.348	0.703
H 32	I often feel hungry	0.404	0.701
H 34	If my meals are delayed I get light-headed	0.453	0.698
SE 14	I often finish my meals quickly	−0.025	0.727
SE 25	I am often last at finishing a meal	0.380	0.701
SE 26	I eat more and more slowly during the course of a meal	0.294	0.707
SE 29	I eat slowly	0.198	0.713
SR 11	I often leave food on my plate at the end of the meal	0.253	0.709
SR 23	I often get full before my meal is finished	0.273	0.708
SR 30	I cannot eat a meal if I have had a snack just before	0.095	0.719
SR 31	I get full up easily	0.225	0.711

**Table 4 nutrients-14-04705-t004:** Promax rotated PCA of AEBQ-Ar.

		Extracted Factors					
		EOE	EF	H	SE	FF	SR
EOE 8	I eat more when I’m worried	0.935					
EOE 5	I eat more when I’m annoyed	0.895					
EOE 10	I eat more when I’m upset	0.864					
EOE 16	I eat more when I’m anxious	0.806					
EOE 21	I eat more when I’m angry	0.768					
EF 3	I enjoy eating		0.907				
EF 1	I love food		0.868				
EF 4	I look forward to mealtimes		0.652				
FF 24	I enjoy a wide variety of foods (Reversed item)		−0.575			0.346	
FR 22	I am always thinking about food		0.416	0.329			
FR 33	When I see or smell food that I like, it makes me want to eat		0.415	0.369			
H 34	If my meals are delayed I get light-headed			0.813			
H28	I often feel so hungry that I have to eat something right away			0.727			
H 9	If I miss a meal I get irritable			0.699			
H 32	I often feel hungry			0.642			
SE 29	I eat slowly				0.853		
SE 26	I eat more and more slowly during the course of a meal				0.848		
SE 14	I often finish my meals quickly (Reversed item)				0.764		
SE 25	I am often last at finishing a meal				0.750		
FF 7	I refuse new foods at first					0.806	
FF 12	I enjoy tasting new foods (Reversed item)					0.777	
FF 19	I am interested in tasting new food I haven’t tasted before (Reversed item)					0.757	
FF 2	I often decide that I don’t like a food, before tasting it					0.545	0.349
SR 23	I often get full before my meal is finished						0.795
SR 31	I get full up easily						0.740
SR 11	I often leave food on my plate at the end of the meal						0.642
SR 30	I cannot eat a meal if I have had a snack just before						0.497

Extraction Method: PCA. KMO = 0.849, Bartelettes sphercity chi-squared test χ^2^ (378) = 6354.8, *p* < 0.001, Determinant Index = 0.001. PCA: Principal Components Analysis; KMO: Kaplan–Meyer–Olkins.

**Table 5 nutrients-14-04705-t005:** A descriptive analysis of AEBQ-Ar subscale scores (factor analysis-based scores).

	Mean (SD) Score *
Hunger (H)	2.82 (0.76)
Enjoyment of Food (EF)	3.36 (0.48)
Emotional Over-Eating (EOE)	2.45 (0.97)
Satiety Response (SR)	3.11 (0.75)
Food Fussiness (FF)	2.72 (0.80)
Slowness in Eating (SE)	2.66 (0.90)

* Maximum possible scores are bounded between 1–5 points.

**Table 6 nutrients-14-04705-t006:** Bivariate correlations between measured AEBQ-Ar scores and other factors.

	H	EF	EOE	SR	FF	SE	Age	BMI
Food approach scales								
Hunger (H)	1							
Enjoyment of Food (EF)	0.461 **	1						
Emotional Over-Eating (EOE)	0.391 **	0.425 **	1					
Food avoidance scales								
Satiety Response (SR)	0.092 *	−0.149 **	−0.129 **	1				
Food Fussiness (FF)	−0.025	−0.123 **	−0.061	0.177 **	1			
Slowness in Eating (SE)	−0.027	−0.057	−0.053	0.243 **	0.035	1		
Participants characteristics								
Age (in years)	−0.171 **	−0.269 **	−0.166 **	0.009	0.005	−0.072	1	
Body Mass Index (BMI)	0.016	0.046	0.155 **	−0.077	0.002	−0.155 **	0.339 **	1
Socioeconomic status index (SESi) score	−0.082 *	−0.084 *	−0.134 **	−0.011	−0.061	−0.136 **	0.464 **	0.184 **

** Correlation is significant at the 0.01 level (2-tailed). * Correlation is significant at the 0.05 level (2-tailed).

## Data Availability

The final version of the AEBQ in Arabic language will be available from the corresponding author (Aldhwayan, M.M.) upon reasonable request.

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
