# Peer review of "The Arabic Version of the Adult Eating Behavior Questionnaire among Saudi Population: Translation and Validation"

_nutrients, 2022, doi:10.3390/nu14214705_

Round 1

Reviewer 1 Report

revise title 

revise keywords 

introduction is too short 

add tables and pictures related 

keep results and discussions together 

recheck whole content flow of information 

The main research is eating behaviour and its relevance in current modern lifestyle. It addresses social issues, and it may be useful. Eating, eating disorders, behaviour and related issues is most common issues and need to be addressed. Methodology should be clearer, add more relevant methods in more details, add related pictures and tables. Conclusions should be improved, make it more concise and informative. The references are appropriate. According to the tables and figures, recheck for typo errors, grammatical errors and add all missing relevant information and details.

Reviewer 2 Report

Dear Authors,

Thank you very much to Editor for inviting me to review your publication. Congratulations on your research on the field of Adult Eating Behavior Questionnaire in the next country (Saudies). Translation and validation of the AEBQ was carried out using similar tools as in other countries. The adaptation of this validated method for Saudi Arabia provides opportunities to use it to assess individuals who require intervention for their weight, eating habits and to identify eating disorders. As the authors themselves write, there are few Arabic tools for assessing eating behavior.

Below are my suggestions / comments:

A pilot study of a 20-person sample was conducted.

Validation of the questionnaire was carried out.

Study sample size calculated properly, but I have doubts that the sample was representative.

Statistical analysis was carried out using multivariate analyses.

Table 1 Households monthly Income Level: required unit of currency.

Incorrectly marked bibliographic items in the text
